# Polyphosphate nanoparticles enhance the fibrin stabilization by histones more efficiently than linear polyphosphates

Miklós Lovas[1], Anna Tanka-Salamon[1], László Beinrohr[1], István Voszka[2], László Szabó[1,3], Kinga Molnár[4], Krasimir Kolev[1] *

1 Department of Biochemistry, Semmelweis University, Budapest, Hungary, 2 Department of Biophysics and Radiation Biology, Semmelweis University, Budapest, Hungary, 3 Department of Functional and Structural Materials, Institute of Materials and Environmental Chemistry, Research Centre for Natural Sciences, Hungarian Academy of Sciences, Budapest, Hungary, 4 Departement of Anatomy, Cell- and Developmental Biology, ELTE Eötvös Loránd University, Budapest, Hungary

* Krasimir.Kolev@eok.sote.hu

## Abstract

### Introduction

Beyond the three-dimensional fibrin network, the mechanical and lytic stability of thrombi is supported by the matrix of neutrophil extracellular traps (NETs) composed of polyanionic DNA meshwork with attached proteins including polycationic histones. Polyphosphates represent another type of polyanions, which in their linear form are known to enhance the fibrin stabilizing effects of DNA and histones. However, *in vivo* polyphosphates are also present in the form of nanoparticles (PolyP-NP), the interference of which with the fibrin/NET matrix is poorly characterized.

### Aims

To compare the effects of linear and nanoparticulate polyphosphates, and their combinations with relevant NET components (DNA, histone H3) on fibrin formation, structure, and lysis in *in vitro* assays focusing on histone-polyphosphate interactions.

### Methods

Transmission electron microscopy and dynamic light scattering for stability of the PolyP-NP preparations. Turbidimetry for kinetics of fibrinogen clotting by thrombin and fibrin dissolution by tissue-type plasminogen activator/plasminogen. Scanning electron microscopy for fibrin structure. Surface plasmon resonance for strength of histone-PolyP interactions.

### Results

Both linear PolyP and PolyP-NP accelerated the fibrin formation and slowed down its dissolution and these effects were strongly dependent on the number of individual PolyP particles and not on their size. Addition of DNA did not modify significantly the PolyP-NP effects on fibrin formation and lysis. Both linear and nanoparticulate PolyP counteracted the effect of

**Data Availability Statement:** All relevant data are within the paper and its Supporting information files.

**Funding:** This study was funded by the Thematic Institutional Excellence funding scheme of the Ministry of Innovation and Technology in Hungary for the Molecular Biology thematic programme of Semmelweis University (https://nkfih.gov.hu/) [TKP2021-EGA-24] and the Hungarian National Research, Development and Innovation Office (https://nkfih.gov.hu/) [137563] in the form of funds to KK, and the ÚNKP Scholarship of the Ministry of Innovation and Technology, Hungary in the form of funds to ML. The funders had no role in study design, data collection and analysis, decision to publish, or preparation of the manuscript.

**Competing interests:** The authors have declared that no competing interests exist.

histone in the acceleration of fibrinogen clotting by thrombin. PolyP-NP, but not linear PolyP enhanced the prolongation of lysis time in fibrin containing histone and caused more pronounced thickening of the fibrin fibers than the linear form. Finally, PolyP-NP bound weaker to histone than the linear form.

## Conclusions

The interaction of PolyP with histone was a stronger modulator of fibrin formation and lysis than its interaction with DNA. In addition, the PolyP nanoparticles enhanced the thrombus stabilizing effects of histone more effectively than linear PolyP.

## Introduction

Neutrophil extracellular traps (NETs) and polyphosphates (PolyPs) are both known as key players of (immuno)thrombosis. Their procoagulant and prothrombotic nature, as well as their interactions in circulation have been recently reviewed in detail [1]. Our previous studies on the effects of NET components on structural and lytic stability of fibrin and plasma clots revealed that DNA, histones and their combinations increase the mechanical stability of clots and hamper tissue-type plasminogen activator (tPA) mediated clot lysis [2,3]. Furthermore, we found, that linear PolyPs stabilize fibrin via size-dependent modulation of its structure and kringle-dependent inhibition of plasmin-mediated fibrinolysis [4], while in combination with NET components, linear PolyPs enhanced the clot stabilizing effects of DNA and histones [5]. In addition to the linear form, nanoparticulate polyphosphates (PolyP-NPs) have also been shown to localize at the surface of activated platelets and activate the intrinsic pathway of coagulation [6], but in contrast to the linear PolyP effects, little is known about their contribution to the stabilization of thrombi by NETs.

In this study, we aimed 1) to develop a new method for preparing stable, pure PolyP-NPs that are suitable for *in vitro* applications to study their role in hemostasis, and 2) to compare and characterize the effects of the linear and nanoparticulate forms of PolyPs and the combined effects of DNA, histones and PolyPs on the kinetics of clotting, structure and lytic susceptibility of fibrin clots *in vitro*.

## Materials and methods

Plasminogen depleted human fibrinogen, alkaline phosphatase, short-chain polyphosphate (PolyP-Lin-45 with an average polymer length of 45 monomers), histones from calf thymus type VIIIS (arginine-rich histone H3) were obtained from Merck Kft. (Budapest, Hungary). Polyphosphate, Medium Chain (PolyP-Lin-100) and Polyphosphate, Long Chain (PolyP-Lin-700) were from Kerafast (Boston, MA, USA). Bovine thrombin was purchased from Serva (Heidelberg, Germany) and further purified by ion-exchange chromatography on sulfopropyl-Sephadex yielding a preparation with a specific activity of 2,100 IU/mg [7] and 1 IU/ml was considered equivalent to approximately 10.7 nM by active site titration [8]. Plasminogen and plasmin were prepared as previously described [9].

### Preparation of PolyP-NPs

Pure PolyP nanoparticles are labile, so we developed a novel technique for preparing stable PolyP nanoparticles based on improvements of earlier work [10,11]. The detailed procedure is

as follows. $Ca^{2+}$-PolyP nanoparticles were generated by adding linear $PolyP_{45}$ (1 mM) to $CaCl_2$ (1.67 mM) in BSA-TBS buffer (8 mM Tris-HCl, pH 9, containing 1 mg/ml bovine serum albumin). $Ca^{2+}$-precipitation was allowed to proceed at room temperature for 20 min under continuous vigorous stirring to achieve homogeneity. Thereafter large aggregates were dispersed by ultrasound sonication with a Branson Sonifier 250 (BRANSON Ultrasonics Corp., Danbury, CT, USA) for 10 min (20 kHz horn frequency, 65 W) with a 5-min pause at half-time. Remnant large aggregates were removed by filtrating them sequentially through a syringe filter of 1.2 μm and then through a filter of 0.22 μm pore size. Finally, the filtrate with the remaining PolyP-NPs was retained and stored at 4˚C.

## Determination of phosphate monomer and polyphosphate nanoparticle concentration

The phosphate monomer concentration of both linear and nanoparticulate polyphosphates was determined by a modified version of the Fiske-Subbarow method [12]. Alkaline phosphatase (20 U/l) was added to polyphosphate solutions and mixtures were incubated for 2 hours at 37˚C to cleave phosphate polymers to monomers. Then a mixture of 4.4%(w/v) ascorbic acid and 4.4%(w/v) trichloroacetic acid (TCA) was added to the samples followed by centrifugation for 2 min at 5,000$g$. The liquid phase was withdrawn and after the addition of 0.2%(w/v) ammonium molybdate the samples were incubated for 1 hour at 50˚C. Following the color reaction, the absorbance of the samples and a dilution series of $NaH_2PO_4$ standard were measured in a CLARIOstar spectrophotometer (BMG Labtech, Ortenberg, Germany) at 700 nm. Particle concentration of the PolyP-NP preparations was estimated considering the phosphate monomer concentration of the samples, the average density of $Ca^{2+}$-PolyP-NPs (3.14 g/cm$^3$) [13] and the radius of the particles calculated according to our transmission electron microscopy and dynamic light scattering measurements using sphere approximation [14]. The preparations of applied polyphosphates are summarized in Table 1.

## Characterization of PolyP-NPs by transmission electron microscopy (TEM)

To visualize and to measure the size of the polyphosphate nanoparticles samples were mounted on Formwar-Carbon coated 300-mesh nickel grids (Nisshin EM Co. Ltd. Tokyo, Japan) and stained with 2%(w/v) uranyl acetate. Samples were examined using a JEOL JEM-1011 electron microscope (JEOL, Ltd., Tokyo, Japan) at an accelerating voltage of 80 kV. Images were taken by Morada 11 Megapixel camera (Olympus, Tokyo, Japan) using iTEM software (Olympus) [15].

**Table 1. Designations of polyphosphates used in this study.**

| Designation | Type and form of polyphosphate |
|---|---|
| PolyP | any type, linear or nanopaticulate |
| PolyP-Lin | any type, linear |
| PolyP-Lin-45 | 45 monomer average size, linear |
| PolyP-Lin-100 | 100 monomer average size, linear |
| PolyP-Lin-700 | 700 monomer average size, linear |
| PolyP-NP | 45 monomer average size, nanoparticulate |
| PolyP-Lin$_m$ | 45 monomer average size, linear at 330 μM monomer equivalent |
| PolyP-Lin$_p$ | 45 monomer average size, linear at $1.96 \times 10^{12}$ particle/l |

## Characterization of PolyP-NPs by dynamic light scattering (DLS)

The stability of polyphosphate nanoparticles was monitored by measuring the size distribution of PolyP-NP samples for 6 days with DLS. Furthermore, the average particle size was checked before each measurement to validate that the nanoparticle diameter is within the target range of 100–200 nm [6]. This was performed using an ALV goniometer with a Melles Griot diode-pumped solid-state laser at 457.5 nm wavelength (type: 58 BLD 301). The intensity of the scattered light was measured at 90˚ and the autocorrelation function was calculated using an IBM PC-based data acquisition system developed in the Department of Biophysics and Radiation Biology of Semmelweis University (Budapest, Hungary). Particle size distributions were determined by the maximum entropy method [16].

## Turbidimetric measurements of fibrin formation and dissolution

In order to maximize the modulator effects we used the highest final PolyP concentration achievable in the reaction mixtures during the turbidimetric measurements which was 330 μM in terms of phosphate monomer that corresponds to $1.96 \times 10^{12}$ PolyP particle/l in terms of particle concentration. PolyPlin concentrations were set to match these values: PolyP-Lin$_p$ samples of $1.96 \times 10^{12}$ PolyP particle/l and PolyP-Lin$_m$ samples of 330 μM monomer concentration were prepared. The equivalence of the concentrations of the three different PolyP solutions is illustrated in S1 Fig.

Formation and dissolution of fibrin clots were followed in 96-well microtiter plates by measuring the light absorbance at 340 nm at 37˚C with a CLARIOstar spectrophotometer (BMG Labtech, Ortenberg, Germany). Fibrinogen (7 μM) containing plasminogen (0.2 μM) was clotted by thrombin (2 nM) with or without the modulators (5 μg/ml DNA, 20 μM histone or the above-mentioned concentrations of PolyPs) and their combinations at 100 μl total volume. Regarding the different $Ca^{2+}$ content of our PolyP-Lin (0 mM) and PolyP-NP preparations (1.67 mM), $Ca^{2+}$-free and $Ca^{2+}$-bearing control samples were used. After turbidity reached the plateau phase 90 μl 140 nM tPA was layered on the surface to trigger lysis, and the measurement was continued until complete dissolution of the clots. Kinetics of clotting and lysis were characterized by the time to reach half-maximal turbidity on the ascending and descending part of the turbidimetric curves (clotting time$_{50}$ and lysis time$_{50}$, respectively).

## Characterization of the structure of fibrin clots by scanning electron microscopy (SEM)

Fibrinogen (7 μM) was clotted by thrombin (2 nM) in the presence of modulators and their combinations at the concentrations used in the turbidimetric measurements. After complete clotting, fibrin was washed 3 times with 100 mM Na-cacodylate pH 7.2 buffer and fixed in 1% (v/v) glutaraldehyde for 30 min, then dehydrated in a series of ethanol dilutions [20–96%(v/v)], 1:1 mixture of 96%(v/v) ethanol/acetone and pure acetone followed by critical point drying with $CO_2$ in an E3000 Critical Point Drying Apparatus (Quorum Technologies, Newhaven, UK). The specimens were then mounted on adhesive carbon discs, sputter-coated with gold in an SC7620 Sputter Coater (Quorum Technologies) and images were taken with scanning electron microscope EVO40 (Carl Zeiss GmbH, Oberkochen, Germany). The images were analyzed to determine the diameter of the fibrin fibers as described previously [2,17].

## Histone-fibrinogen gelation

Previous studies demonstrated the gelation of histone-fibrinogen mixtures and its consequences on maximal turbidity during clotting assays [18]. We investigated the impact of this phenomenon to avoid misleading interpretations of our results with histones.

Fibrinogen at 7 µM was mixed with histones at various concentrations (in the range of 0–26 µM) in the presence or absence of 50 nM plasmin in 96-well titration plates and the turbidity of the samples was monitored by a CLARIOstar spectrophotometer at 340 nm for 150 min at 37˚C. Samples (10 µl) were withdrawn at 0, 60 and 150 minutes and treated with 90 µl 1%(w/v) sodium dodecyl sulfate (SDS) in 100 mM Tris-HCl, 100 mM NaCl, pH 8.2 and polyacrylamide gel electrophoresis (PAGE) was performed on a 4–15% gradient gel followed by silver staining.

## Surface plasmon resonance (SPR) measurements of histone-PolyP interactions

The interaction of histone and PolyP was evaluated with surface plasmon resonance (SPR) technology on SR7500DC system (Reichert Technologies Life Sciences, NY, USA) with standard software SPRAutolink 1.1.14-T provided by the manufacturer for reporting and analyzing data. The running buffer was 10 mM HEPES-NaOH 150 mM NaCl pH 7.4 containing Tween-20 at 0.005 w/w% (HBS-T) in all experiments, except when PolyP-NP was measured, where 1.67 mM $CaCl_2$ was also added. One of the channels on a dextran coated gold chip (Reichert SR7000 chip, 500 kDa carboxymethyl-dextran, part. no. 13206066) was coated non-covalently with 0.25 mg/ml Histone H3 in 10 mM Na-acetate pH 5.5 at a flow rate of 25 µl/min for 7 min [19]. After the coating, the chip was washed and equilibrated with the running buffer for ~10 min. For the interaction measurements 1 mM PolyP solutions were injected at 25 µl/min flow rate for 7 min. Results are reported in relative units (RU) as a difference in the signal between the histone and the reference channels. After each measurement, the chip surface was stripped and regenerated using 6 M guanidine hydrochloride in unbuffered ultrapure water, injected at 25 µl/min for 7 min. Guanidine was chosen to prevent precipitation of $Ca^{2+}$ salts that is observed with more common stripping methods. The system was then washed with pure water and re-equilibrated with running buffer for a subsequent non-covalent coating step.

## Statistical analysis

The distribution of the data on fiber diameter was analyzed using an algorithm described previously: theoretical distributions were fitted to the empirical data sets and compared using Kuiper's test and Monte Carlo simulation procedures running under MatlabR2018b (The Mathworks, Natick, MA) [20]. Kolmogorov-Smirnov test was used for statistical evaluation of differences in distribution of other experimental datasets in this report (Statistics and Machine Learning Toolbox v.11.4 of MatlabR2018b) at $p<0.05$.

## Results

### Generation of stable polyphosphate nanoparticles

First, we aimed to develop a novel technique for preparing stable PolyP nanoparticles (PolyP-NP). Our method was based on previous work by Morrissey et al. with further modifications [10]. The structure and stability of the generated nanoparticles were investigated by TEM and DLS. According to TEM micrographs small, electron-dense $Ca^{2+}$-polyphosphate nanoparticles were formed with diameters in the range of 80–150 nm (Fig 1A). The size of the particles according to DLS was similar: peak volume probabilities were observed at 110–200 nm particle size (Fig 1B). The slightly larger size values with DLS were probably due to differences in the hydration state of the samples: for TEM the samples were dehydrated, while in DLS samples preserved their hydration shell. The nanoparticle stability was monitored by measurements of the size distribution in the PolyP-NP preparation with DLS. Till the 6[th]

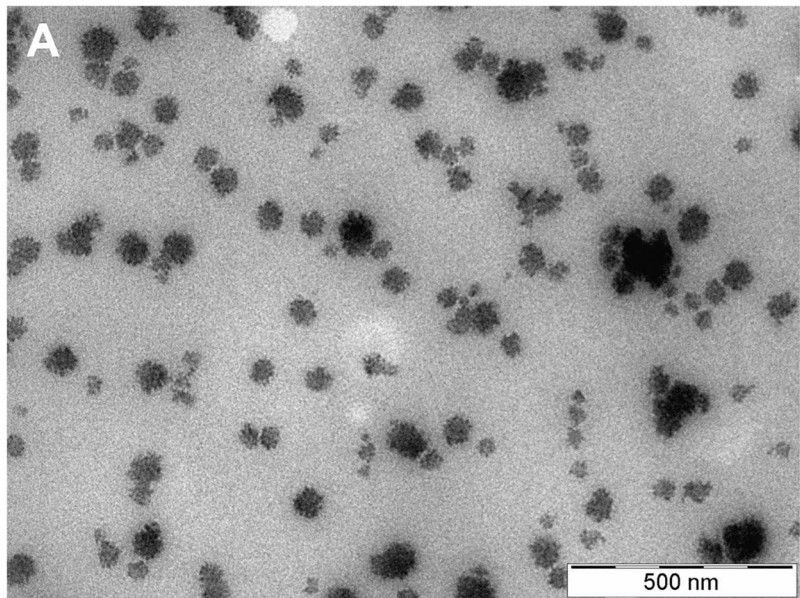

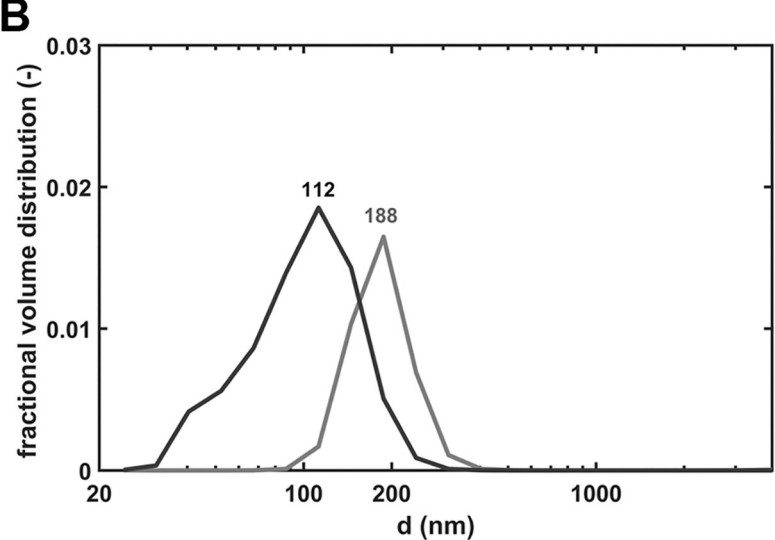

**Fig 1. Structure and size distribution of polyphosphate nanoparticles (PolyP-NPs).** PolyP-NPs were formed by precipitation of soluble PolyP-Lin-45 (1 mM) with $CaCl_2$ (1.67 mM) in BSA-TBS, then sonicated and filtered through a syringe filter with a pore size of 220 nm. *Panel A*: Transmission electron micrographs of PolyP-NP (TEM). *Panel B*: Fractional volume distribution of PolyP-NPs according to their particle size in PolyP-NP preparation examined by dynamic light scattering (DLS). The distribution curves of two different PolyP-NP preparations are shown to illustrate the minimal and maximal median values of particle size in the PolyP-NP preparations used in this study.

storage day no change in the median diameter of PolyP-NPs was observed suggesting that neither aggregation, nor disintegration of the PolyP-NPs occurred in the preparations for at least 5 days stored in $CaCl_2$-BSA-TBS at 4°C (Fig 2).

## Fibrin formation and lysis in the presence of polyphosphates

All examined types of polyphosphate preparations (PolyP-Lin$_m$, PolyP-NP, PolyP-Lin$_p$) decreased the clotting time$_{50}$ and increased the lysis time$_{50}$, but the size of their effect differed

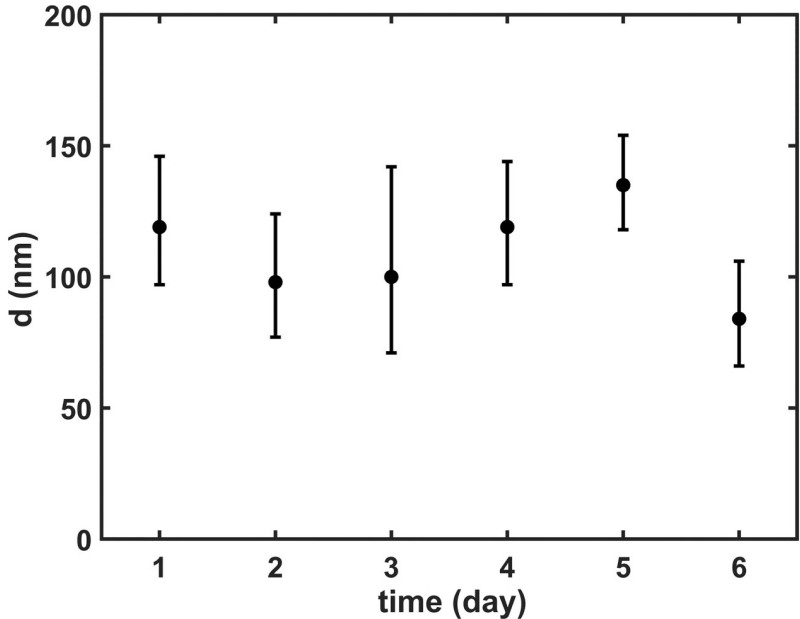

**Fig 2. Stability of polyphosphate nanoparticles (PolyP-NPs).** The particle size of the PolyP-NPs was monitored by dynamic light scattering (DLS) for 6 days as a measure of the preparation stability. Median particle diameters and interquartile range are shown (n = 100).

(Table 2). Despite the difference in particle size (large aggregates in PolyP-NP and isolated molecules of 45 monomers in PolyP-Lin$_p$) the magnitude of the effect on the time course of clotting and lysis was similar for the preparations with identical number of particles (PolyP-NP and PolyP-Lin$_p$; 19 and 12% faster clotting, 1.76-, and 1.28-fold slower lysis, respectively, compared to the PolyP-free control). However, particles of the same structure exerted stronger effects at higher numbers (PolyP-Lin$_m$ vs. PolyP-Lin$_p$; 45 and 12% faster clotting, 3.08-, and 1.28-fold slower lysis, respectively). These data show that the modulation of the kinetics of fibrin formation and dissolution depends primarily on the number of individual PolyP particles and not on their size.

**Table 2. Effects of different types of polyphosphates on fibrin formation and clot lysis.**

| | | No additive | Ca$^{2+}$ | PolyP-Lin$_m$ | PolyP-Lin$_p$ | PolyP-NP |
|---|---|---|---|---|---|---|
| clotting time$_{50}$ (min) | median | **52.5** | **38.56** | **28.62** | **46.17** | **31.11** |
| | interquartile range | 14.28 | 8.75 | 6.43 | 5.46 | 7.27 |
| | $P$-value (vs. no additive) | - | $3.30\times10^{-8}$ | $1.08\times10^{-14}$ | $7.60\times10^{-3}$ | - |
| | $P$-value (vs. Ca$^{2+}$) | $3.30\times10^{-8}$ | - | - | - | $1.24\times10^{-5}$ |
| lysis time$_{50}$ (min) | median | **12.01** | **17.21** | **37.03** | **15.39** | **30.32** |
| | interquartile range | 6.28 | 11.79 | 5.52 | 9.27 | 8.31 |
| | $P$-value (vs. no additive) | - | $1.30\times10^{-3}$ | $2.45\times10^{-17}$ | 0.27 | - |
| | $P$-value (vs. Ca$^{2+}$) | $1.30\times10^{-3}$ | - | - | - | $5.64\times10^{-17}$ |

Mixture of fibrinogen (7 µM) and plasminogen (0.2 µM) was clotted by thrombin (2 nM) in the presence or absence of linear PolyP (PolyP-Lin$_m$ or PolyP-Lin$_p$, 330 µM and $1.96\times10^{12}$ particle/l, respectively) or PolyP-NP (330 µM which corresponds to $1.96\times10^{12}$ particle/l) and the turbidity was monitored at 340 nm. At the plateau phase tPA was added to the surface of the clots and measurement was continued until complete lysis. The time corresponding to half maximal turbidity was used as a measure of reaction kinetics during both clotting and lysis (clotting-, and lysis time$_{50}$). $P$-values are shown according to Kolmogorov-Smirnov test in comparison with the relevant control (n = 20).

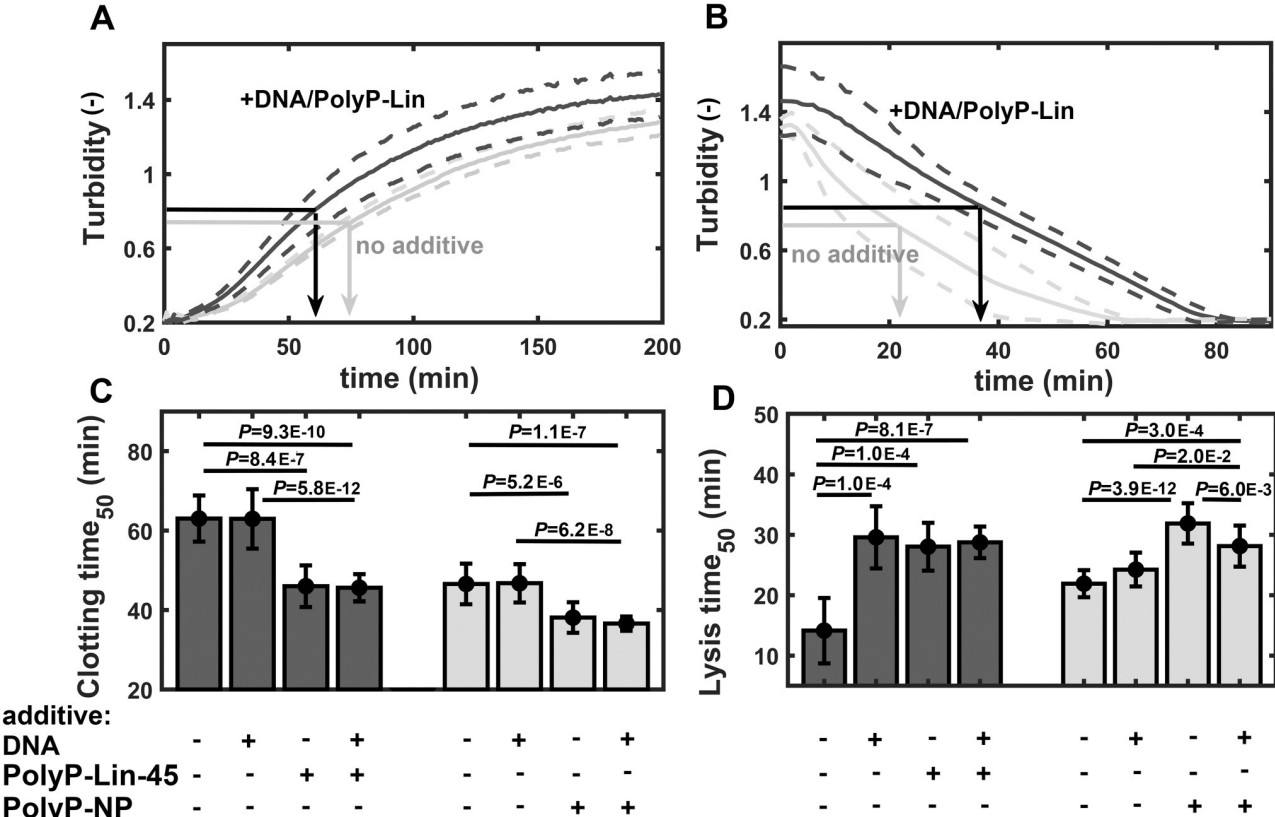

**Fig 3. Combined effects of DNA and polyphosphates on fibrin formation and lysis.** Fibrinogen (7 μM) containing plasminogen (0.2 μM) was clotted by thrombin (2 nM) in the presence or absence of DNA (50 μg/ml), linear PolyP (330 μM) or PolyP-NP (330 μM) and the absorbance at 340 nm (Turbidity) was measured (Panel A). At the plateau phase tPA (140 nM) was layered on the surface of clots and measurement was continued until complete lysis (Panel B). Arrows indicate the time to reach half-maximal turbidity (clotting time$_{50}$ and lysis time$_{50}$) in the ascending and descending phase of the representative turbidimetric curves in panels A and B, respectively. Median values and interquartile range (error bars) of clotting time$_{50}$ (Panel C) and lysis time$_{50}$ (Panel D) are shown in black (no Ca$^{2+}$) or grey (1.67 mM Ca$^{2+}$). *P*-values for differences of the indicated pairs of datasets are shown according to Kolmogorov-Smirnov test (n = 25).

## Combined effects of DNA and polyphosphates on fibrin formation and lysis

*In vivo* blood clots are known to contain polyanions (e.g. PolyPs and DNA) and their interplay as modulators of blood clotting and lysis is potentially relevant and partially revealed [4,21]. According to our turbidimetry data, the presence of DNA neither altered clotting time$_{50}$ alone, nor modulated the enhancement of clotting by PolyPs (Fig 3A and 3C). In the fibrinolytic assay DNA behaved in a Ca$^{2+}$-dependent manner. In a Ca$^{2+}$-free milieu, DNA doubled the lysis time$_{50}$ similarly to PolyP-Lin$_m$, but in combination the two polyanions did not show any additivity of their effects (Fig 3B and 3D). In the presence of Ca$^{2+}$, the lysis was not affected by DNA, while PolyP-NP increased lysis time$_{50}$ by 45% and this anti-fibrinolytic effect was only slightly moderated by DNA.

## Combined effects of histones and polyphosphates on fibrin formation and lysis

Similarly to extracellular DNA, histones play critical roles in host defense, exerting procoagulant, proinflammatory and antifibrinolytic effects (reviewed in [22]). Their known impact on

**Table 3. Effects of histone on fibrin formation and lysis.**

| | | No additive | Histone 3.5 µM | Histone 7 µM | Histone 13 µM | Histone 20 µM |
|---|---|---|---|---|---|---|
| clotting time$_{50}$ (min) | median | **53.63** | **35.65** | **30.68** | **22.50** | **11.08** |
| | interquartile range | 15.26 | 8.20 | 4.28 | 9.68 | 7.79 |
| | *P*-value | - | $8.71 \times 10^{-5}$ | $2.86 \times 10^{-7}$ | $2.86 \times 10^{-7}$ | $2.86 \times 10^{-7}$ |
| lysis time$_{50}$ (min) | median | **14.62** | **31.18** | **40.00** | **44.82** | **59.95** |
| | interquartile range | 6.10 | 16.13 | 24.72 | 16.78 | 6.78 |
| | *P*-value | - | $1.10 \times 10^{-3}$ | $1.70 \times 10^{-4}$ | $3.99 \times 10^{-6}$ | $3.28 \times 10^{-6}$ |

Fibrinogen (7 µM) containing plasminogen (0.2 µM) was clotted by thrombin (2 nM) in the presence of histone at various concentrations. Reaction kinetics was monitored and lysis induced as in Fig 3. *P*-values are shown according to Kolmogorov-Smirnov test in comparison with no additive (n = 15).

clotting and lysis kinetics was confirmed by our measurements showing a concentration-dependent decrease in clotting-, and prolongation of lysis times (Table 3). The plausible, electric charge-dependent interplay of histones and polyphosphates during *in vivo* thrombus formation justifies comparative evaluation of the effect of histones with the histone+PolyP combinations on the kinetics of clotting and lysis.

The effect of histone on clotting time (clotting time$_{50}$ decreased by 79% in a Ca$^{2+}$-free and by 73% in a Ca$^{2+}$-bearing milieu) was opposed by all of the polyphosphates: histone+-PolyP-Lin$_m$, histone+PolyP-Lin$_p$ and histone+PolyP-NP combinations increased clotting time$_{50}$ 3.3-, 2.5- and 1.6-fold, respectively, compared to histone's effect alone (Fig 4A and 4C). Histone's effect on lysis time$_{50}$ (a 4.1-fold increase in the absence of Ca$^{2+}$ and a 2.5-fold increase in the presence of Ca$^{2+}$) was not influenced by histone+PolyP-Lin$_m$ or histone+-PolyP-Lin$_p$ combinations, but it was enhanced by histone+PolyP-NP (Fig 4B and 4D). The PolyP-NP's effect on lysis time$_{50}$ (increase by 76%) was remarkably strengthened by histone (increased further by 118%). According to these findings we can conclude, that 1) in fibrin polymerization all types of polyphosphates oppose the accelerating effect of histone, but the histone effect is least hampered by PolyP-NP; 2) in fibrinolysis only the polyphosphate in nano-particulate form (and not the linear one) acts in synergism with histones to inhibit the process.

## Histone-fibrinogen aggregation and its effect on fibrinogen digestion by plasmin

During the investigation of the effects of histone alone or in combination with PolyPs on fibrinogen clotting and lysis, we observed increased turbidity at the onset of clot formation (Fig 4A) and prolonged fibrin lysis time (Table 3, Fig 4D). Our results confirmed the earlier findings of histone-fibrinogen gelation: incubating fibrinogen with histone resulted in increasing histone-fibrinogen aggregation (Fig 5A), and fibrinogen became resistant to lysis by plasmin (Fig 5B) [18].

## Combined effects of histones and polyphosphates on fibrin structure

The kinetics of fibrin formation are known to affect the ultimate clot architecture (fiber size, branching density, porosity), whereas the fibrin structure is an essential determinant of fibrinolysis (reviewed in [23]). According to the results reported in the preceding sections, histones (and not DNA) modulated significantly the polyphosphate effects on fibrin formation and lysis. Thus, modification of the fibrin structure in the presence of polyphosphates in combination with histones could be hypothesized as a mechanism underlying our current findings. We characterized one parameter of the fibrin structure—the fiber thickness. In line with our earlier

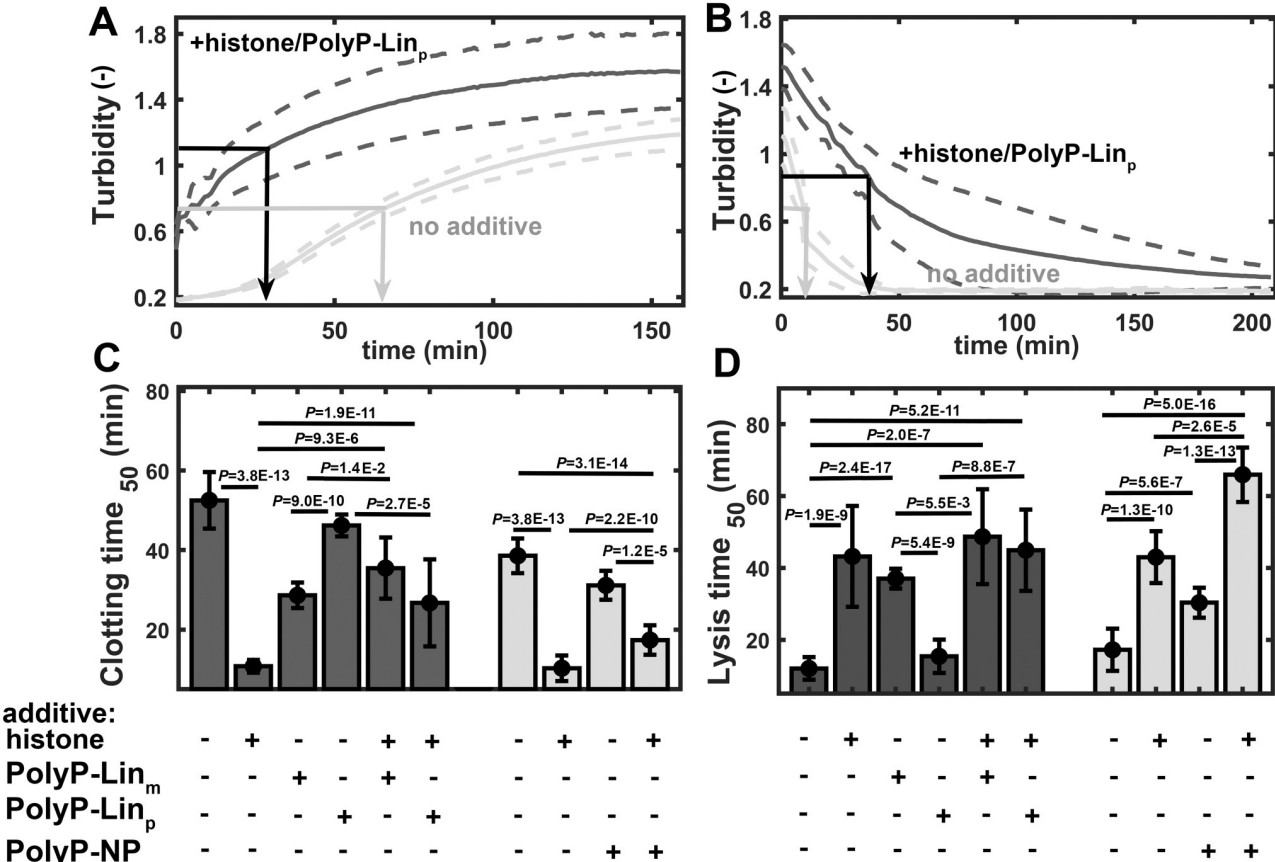

**Fig 4. Combined effects of histone and polyphosphates on fibrin formation and lysis.** Fibrinogen (7 μM) mixed with plasminogen (0.2 μM) was clotted by thrombin (2 nM) in the presence or absence of histone (20 μM), linear PolyP-Lin$_m$ (330 μM), linear PolyP-Lin$_p$ ($1.96 \times 10^{12}$ particle/l equivalent to 330 μM) or nanoparticulate ($1.96 \times 10^{12}$ particle/l) polyphosphate and the absorbance at 340 nm (Turbidity) was measured (Panel A). At plateau phase tPA (140 nM) was layered on the surface of clots and measurement was continued until complete lysis (Panel B). Arrows indicate the time to reach half-maximal turbidity (clotting time$_{50}$ and lysis time$_{50}$) in the ascending and descending phase of the representative turbidimetric curves in panels A and B, respectively. Median values and interquartile range (error bars) of clotting time$_{50}$ (Panel C) and lysis time$_{50}$ (Panel D) are shown in black (no Ca$^{2+}$) or grey (1.67 mM Ca$^{2+}$). *P*-values for differences of the indicated pairs of datasets are shown according to Kolmogorov-Smirnov test (n = 15).

results [2], histone itself increased the median fibrin fiber diameter by 19% (Fig 6C). Poly-P-Lin$_m$ and its combination with histone increased the fiber diameter by 13% and 23%, respectively (not shown). Ca$^{2+}$ itself reduced fiber diameter by 11%, while PolyP-NP and histone +PolyP-NP combination increased it by 23% and 28%, respectively. In summary, the examined modulators and their combinations all increased the fibrin fiber diameter, however to a different extent and the combination of a polyanionic modulator (polyphosphates in different form) and a polycationic modulator (histone) did not neutralize, but rather reinforced their standalone effects.

## Histone-PolyP binding

We recorded the SPR sensorgrams of four different polyphosphate forms (nanoparticulate form and linear variants of three differently lengths) flowing over histone non-covalently immobilized on carboxymethyl-dextran surface in the SPR cell (Fig 7). We employed a non-covalent immobilization strategy to prevent chemical modifications of the histone that could

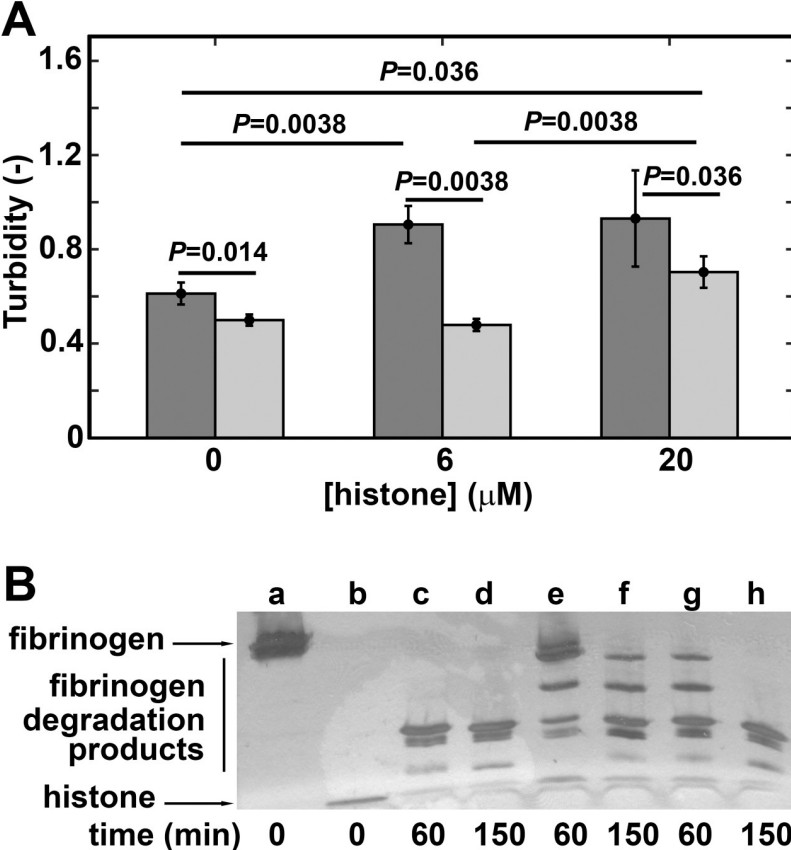

**Fig 5. Histone-fibrinogen aggregation and its effect on fibrinogen digestion by plasmin.** Fibrinogen solution (7 μM) was added to histone (0, 6 or 20 μM) in the absence (black bars in *panel A*, a-b lanes in *panel B*) or presence (grey bars in *panel A*, c-h lanes of *panel B*) of plasmin (50 nM). *Panel A*: Aggregation was monitored by turbidimetry for 150 min. Mean maximal turbidity ± SD and *P*-values according to Kolmogorov-Smirnov test for differences between the indicated pairs of datasets are shown (n = 15). *Panel B*: Digestion of fibrinogen-histone mixtures by plasmin. Non-reducing samples were taken for SDS-PAGE at 0, 60 or 150 min and electrophoresis was performed on a 4–15% polyacrylamide gel followed by silver staining. *Panel B*: *a*, *c*, *d*: Fibrinogen; *b*: Histone (20 μM); *e*, *f*: Fibrinogen + histone (20 μM); *g*, *h*: Fibrinogen + histone (6 μM).

change the positive charge of its lysine and arginine residues conceivable in the targeted inter-actions (as described elsewhere, [19]). This strategy also circumvented the potential problem of histone denaturation, as each run started with freshly immobilized histone material. Poly-phosphates were applied at concentrations spanning the range 0.1–1 mM (concentration in monomeric phosphate equivalents) (S1 File). We found that the nanoparticle form interacted weakly with histone with estimated dissociation equilibrium constant in the millimolar range (S1 File), whereas the linear polyphosphate forms resulted in a decline in the signal (Fig 7). This latter finding is consistent with a strong linear polyphosphate-histone binding that par-tially stripped the surface of non-covalently attached histone. There was no significant differ-ence between the linear polyphosphate forms of different size (45, 100 or 700 monomeric units) at equivalent monomeric phosphate concentrations. To sum up, all forms of polypho-sphate were able to bind histone, but with a marked qualitative difference in the strength of binding of nanoparticulate and linear forms. The stripping effect could be of interest, if poly-phosphates and histones interact under flow conditions *in vivo*.

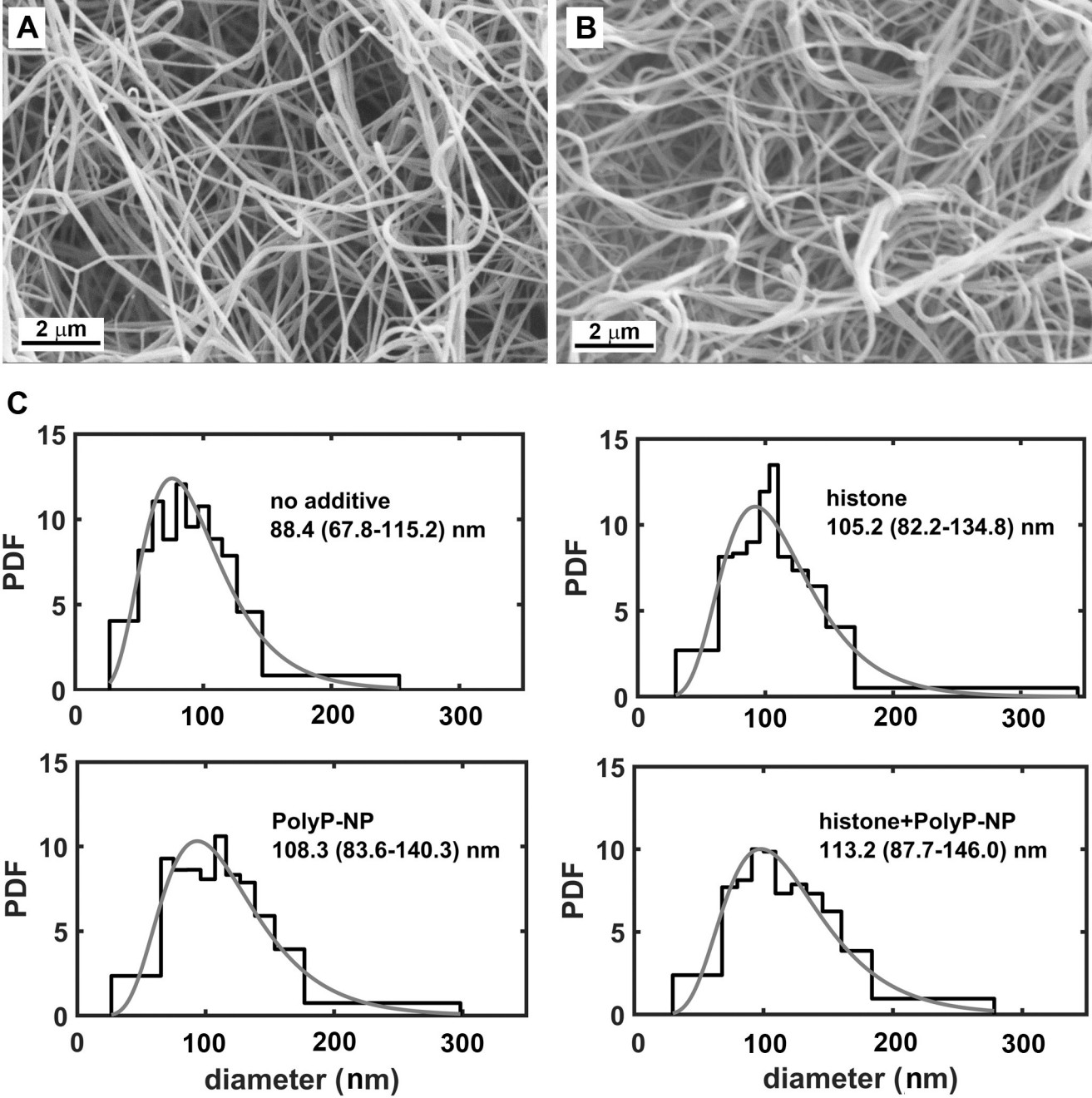

**Fig 6. Effects of histone and PolyP-NP on clot structure.** Representative scanning electron micrographs of fibrin prepared in the absence (*panel A*) or presence of modulators (histone+PolyP-NP (*panel B*), histone or PolyP-NP) as described in Materials and methods. Panel C: The diameters of 300 fibrin fibers were measured from each of 3 independent fibrin sample images and the probability density function (PDF) of the diameter values is shown. The empirical data are shown as a histogram with bins of equal area (each bin has equal product of density and span of data), whereas the grey line indicates the lognormal theoretical distribution that fitted best these data. All differences in the median fiber diameters were significant at $p < 0.05$, according to Kuipers's test of the distributions [20].

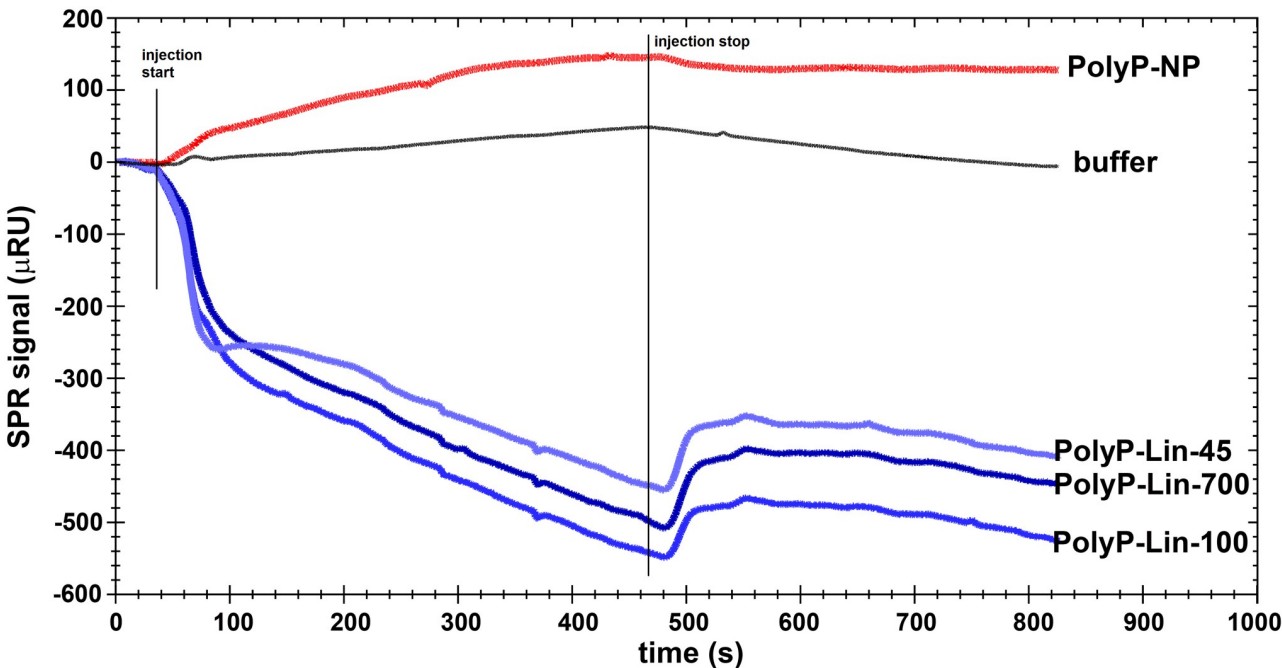

**Fig 7. Representative SPR sensorgrams of the interaction of different polyphosphate forms and histone.** Histone, non-covalently attached to a carboxymethyl-dextran surface, was exposed to a flow of polyphosphate of the indicated type at 1 mM concentration as described in Materials and methods. The nanoparticulate form of polyphosphate shows a small, but positive SPR signal, indicative of an ordinary weak interaction. In contrast, the linear polyphosphate forms strip the non-covalently bound histone from the chip, irrespective of their length.

## Discussion

Neutrophil extracellular traps were first described as novel players of the innate immune system against bacterial infection [24]. In addition to NETosis, neutrophils contribute to the localization of microbial infection by promoting blood coagulation [21]. Furthermore, there is growing evidence that NETs are key players in the formation of pathological thrombi [25]. In our previous works we focused on the effects of NET components (histones and DNA) on thrombus formation, structure and stability [2,3,26]. Besides DNA, another biorelevant, negatively charged polyanion, polyphosphate emerged as a potential modulator of thrombus formation and clot lysis during the last decade [1]. Polyphosphates are stored within the cells in the presence of divalent metal ions and upon platelet activation $Ca^{2+}$-precipitated PolyPs are released in addition to the soluble, linear form. PolyP-NPs are proven to associate to platelet membrane and to be more potent activators of the contact pathway of blood coagulation than the linear form [6]. This finding raised the need to compare the effects of these two forms of polyphosphates on thrombus formation. Multifaceted interactions of linear polyphosphates with thrombogenic factors have been observed. Linear polyphosphates accelerate blood clotting by activating the contact pathway and activation of factor V, which counteracts the action of a major anticoagulant protein, tissue factor pathway inhibitor [27]. They also delay clot lysis through enhancing the action of thrombin-activatable fibrinolysis inhibitor [27]. Linear polyphosphates induce directly platelet aggregation and the long-chain variants are more potent in this action [28]. Linear polyphosphates modulate platelet activation indirectly, too. They induce von Willebrand factor release from endothelial cells through a high mobility group box 1 dependent mechanism resulting in platelet string formation [29]. Furthermore, the coexistence of neutrophils and platelets at the site of (immune)thrombosis allows for interplay

between NET components and polyphosphates [1]. However, linear polyphosphates are unstable in circulation because of the presence of phosphatases. Thus, it is important to assess, if the particulate form of the polyphosphates that is more resistant to degradation can exert the same prothrombotic effects as their linear counterpart. To this end we produced a stable suspension of polyphosphate nanoparticles and in the present study we focused on one aspect of the multifaceted role of polyphosphates in thrombus formation, structure and lysis. We compared the effects of linear and nanoparticulate polyphosphates alone and in combination with NET components (DNA, histone) utilizing a simplified thrombin-mediated fibrin clot model.

## Changes in the kinetics of clotting and lysis: The role of the nanoparticulate form emerges

When polyphosphates acted alone or in combination with DNA the linear form exerted the most pronounced effect on both clotting and lysis time$_{50}$ values. According to our present observations the main determinant of these PolyP effects was the number of particles and DNA did not significantly modify the PolyP effects on the clotting and lysis kinetics. However, when acting together with histone, a new perspective emerged. Firstly, histone and PolyPs modified each other's effects during both clotting and lysis. Secondly, histone exerted its effects on PolyP-Lin$_m$ and PolyP-NP in a different way: the former was hindered (during clotting) or slightly enhanced (during lysis), while the latter was remarkably reinforced during both clot formation and lysis. Based on these findings we conclude that: 1) differences in the intensity of interactions between DNA-PolyPs and histone-PolyPs can be attributed to the overall charges of the molecules: opposite charges in the case of histones and PolyPs are accompanied by stronger interactions; 2) the disparity of the histone-PolyP-Lin$_m$ and histone-PolyP-NP interplay in fibrin turnover suggests differences in their molecular interactions. The latter interpretation was confirmed by SPR measurements. Both linear (PolyP-Lin$_m$) and nanoparticulate PolyP bound to histone, but their SPR signals diverged suggesting that through stronger binding to the histone, the linear variants competed successfully with the non-covalent immobilization forces of the carboxymethyl-dextran matrix, whereas the weaker interactions of the PolyP-NP were not able to detach the complexes from the solid support. An important *in vivo* implication of this finding could be a more pronounced procoagulant and antifibrinolytic effects of polyphosphates in nanoparticulate form in extracellular compartments, where they are retained by the immobilized histone (e.g. in the meshwork of NETs) and enhance the prothrombotic effect of the latter in contrast to the linear alternates, which would rather contribute to the removal of the histone under flow.

## Modification of fibrin structure: A plausible mechanism for the combined effects of histone and polyphosphates on the lytic stability of composite clots

According to earlier studies with pure fibrin, accelerated clot formation (e.g. by higher thrombin concentration) results in thinner fibers [30–32]. In our current study the fibrin fiber diameter was increased by histone, polyphosphates and their combinations despite accelerated clotting phase. Our earlier studies with composite clots have clarified the ultrastructural background of these effects using a small-angle X-ray scattering (SAXS) technique [2]. We have demonstrated that both histone and DNA cause fibrin fiber thickening based on distinct molecular mechanisms: histone interferes with the lateral organization of protofibrils, resulting in lower protofibril density and thicker fibers, while DNA preserves the protofibril-to-protofibril distance, but disrupts the regular longitudinal alignment of the fibrin monomers [2]. As for PolyP, it was shown earlier by others, that the linear form incorporates into polymerizing

fibrin clots [33], which may be the reason for its fiber thickening effect. We have recently shown that linear polyphosphates enhance the fiber-thickening effect of histones [5], and our current results provide evidence that in the form of nanoparticles they are even stronger structural modifiers.

The consequences of the changes of fiber size in terms of lytic susceptibility of fibrin depend on the cause of this structural modification. If higher thrombin activity generates fibrin monomers at higher rate, their fast polymerization results in densely branched fibrin meshwork composed of thin fibers that is resistant to lysis [34, reviewed in 35]. However, if the cause of changes in fibrin architecture is incorporation of modifier molecules, the thicker fibers are more resistant to lysis. Because the structure of the major fibrin-degrading protease, plasmin is optimal for the inter-protofibril distances within modifier-free fibrin fibers (reviewed in [23]), looser packaging of the protofibrils in the fibers would decrease its catalytic efficiency, as we have previously described for the breakdown of fibrin containing histones [2] or combination of histone and linear polyphosphate [5]. This interpretation is supported by the current result that the nanoparticulate PolyP with the strongest steric effect on the fiber geometry enhances stronger the anti-fibrinolytic action of histones than the linear variants. The known anti-fibrinolytic effect of histone-induced aggregation of fibrinogen [36] was observed in our experimental setting too, and could contribute as an independent factor to the lytic resistance of the composite clots.

### Introducing a novel method to prepare stable PolyP-NPs

In recent years, different techniques were developed for generating stable nanoparticulate PolyPs. Colloidal confinement of PolyP on gold or attaching PolyP to silica nanoparticles resulted in enhanced procoagulant function with stability for even 24 weeks, thus potential candidates were developed for the treatment of internal injuries or a range of other haemorrhagic scenarios [37,38]. However, there is limited experience with generating pure PolyP-NPs, moreover, developed particles are stable only for a few hours [10]. Our paper presents a further improved version of previously described method for preparing pure PolyP-NPs [10,11]. The generated particles were of a size similar to those secreted by platelets *in vivo* (100–200 nm [6]), retained the former introduced polyphosphate functions on clot forming and lysis and were stable for at least 5 days stored in BSA-TBS at 4˚C.

### Conclusions

In the past decade, the role of a secondary scaffold formed by activated neutrophil granulocytes has been conceived as an essential factor in thrombus evolution beyond the formation of a 3-dimensional fibrin network. The main effector of NETs, histone possesses several cytotoxic and prothrombotic effects and its overall impact is modified by polyanions (DNA, heparin derivatives and polyphosphates) in the microenvironment of clots. Heparinoids counteract, while DNA and polyphosphates enhance the clot stabilizing effects of histone. The focus of the present work was the comparison of the effects of linear and nanoparticulate polyphosphates alone and in combination with relevant NET components on thrombus formation, structure, and lysis. We found that the PolyP-histone interaction is more significant than the PolyP-DNA interplay probably due to differences in molecular charges: opposite charges are accompanied by stronger interactions. In addition, PolyP nanoparticles enhance the thrombus stabilizing effects of histone more effectively than the linear form. The underlying mechanism of this difference could be related to the weaker PolyP-NP/histone binding observed by us. The weak binding of PolyP-NP to histones allows for independent action of these fibrin-stabilizing factors with consequent additivity of their effects on fibrin fiber thickening and lytic resistance.

In contrast, the stronger linear PolyP-histone interaction counteracts more efficiently the accelerating effect of histones in clotting, but their complex formation abrogates the independence of their individual effects on fibrin structure and lysis and thus their combined action is not additive in the stabilization of fibrin. Therefore, nanoparticulate PolyP may be a relevant target of antithrombotic or thrombolytic agents and the novel method presented here for preparing stable PolyP-NPs may serve this line of investigation.

## Supporting information

**S1 Fig. Relation of the monomeric and particular concentrations of the PolyP preparations.**
(TIF)

**S1 File. Original SPR signal data for the interaction of histone and polyphosphates in nanoparticulate and linear form.**
(ZIP)

## Acknowledgments

The authors are grateful to Györgyi Oravecz and Krisztián Bálint for excellent technical assistance.

## Author Contributions

**Conceptualization:** Miklós Lovas, Anna Tanka-Salamon, Krasimir Kolev.

**Data curation:** Miklós Lovas, Anna Tanka-Salamon, László Beinrohr, István Voszka.

**Formal analysis:** Anna Tanka-Salamon, Krasimir Kolev.

**Funding acquisition:** Miklós Lovas, Krasimir Kolev.

**Investigation:** Miklós Lovas, Anna Tanka-Salamon, László Beinrohr, István Voszka, László Szabó, Kinga Molnár.

**Methodology:** Anna Tanka-Salamon, László Beinrohr, István Voszka, László Szabó, Kinga Molnár, Krasimir Kolev.

**Project administration:** Krasimir Kolev.

**Resources:** Miklós Lovas, Krasimir Kolev.

**Software:** László Beinrohr, István Voszka.

**Supervision:** Krasimir Kolev.

**Validation:** Miklós Lovas, Anna Tanka-Salamon, László Beinrohr, Krasimir Kolev.

**Visualization:** Miklós Lovas, Anna Tanka-Salamon, László Beinrohr, László Szabó, Kinga Molnár, Krasimir Kolev.

**Writing – original draft:** Miklós Lovas, Anna Tanka-Salamon.

**Writing – review & editing:** Miklós Lovas, László Beinrohr, István Voszka, László Szabó, Kinga Molnár, Krasimir Kolev.

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
