## [Decision Letter · Decision Letter 0]

28 Jan 2022

PONE-D-21-40112Polyphosphate nanoparticles enhance the fibrin stabilization by histones more efficiently than linear polyphosphatesPLOS ONE

Dear Dr.Kolve,

Thank you for submitting your manuscript to PLOS ONE. After careful consideration, we feel that it has merit but does not fully meet PLOS ONE’s publication criteria as it currently stands. Therefore, we invite you to submit a revised version of the manuscript that addresses the points raised during the review process. Please response all the comments/questions raised by the reviwers point to point with evidance as to convince the reviwers scientifically.

We look forward to receiving your revised manuscript.

Kind regards,

Gausal A. Khan, Ph.D;CSci,FRSB

Academic Editor

PLOS ONE

Journal Requirements:

[The authors are grateful to Györgyi Oravecz and Krisztián Bálint for excellent technical assistance. This work was supported by the Thematic Institutional Excellence Programme of the Ministry of Human Capacities in Hungary for the Molecular Biology thematic programme of Semmelweis University (TKP2021-EGA-24) and by the Hungarian National Research, Development and Innovation Office (NKFIH) #137563. ML was supported by the ÚNKP Scholarship of the Ministry of Innovation and Technology, Hungary.]

 [KK: Thematic Institutional Excellence Programme of the Ministry of Human Capacities in Hungary for the Molecular Biology thematic programme of Semmelweis University (TKP2021-EGA-24) https://nkfih.gov.hu/

KK: Hungarian National Research, Development and Innovation Office (NKFIH) #137563 https://nkfih.gov.hu/

The funders had no role in study design, data collection and analysis, decision to publish, or preparation of the manuscript.]

Additional Editor Comments:

All the comments /questions raised by the reviwers should be addressed point to point with experimental evidances.

Reviewers' comments:

Reviewer's Responses to Questions

**Comments to the Author**

1. Is the manuscript technically sound, and do the data support the conclusions?

Reviewer #1: Yes

Reviewer #2: Partly

2. Has the statistical analysis been performed appropriately and rigorously? 

Reviewer #1: Yes

Reviewer #2: Yes

3. Have the authors made all data underlying the findings in their manuscript fully available?

Reviewer #1: Yes

Reviewer #2: Yes

4. Is the manuscript presented in an intelligible fashion and written in standard English?

Reviewer #1: Yes

Reviewer #2: Yes

5. Review Comments to the Author

Reviewer #1: In this manuscript “ Polyphosphate nanoparticles enhance the fibrin stabilization by histones more efficiently than linear polyphosphates” Lovas et al demonstrated Polyphosphate nanoparticle with histone is the stronger modulator of fibrin formation than the linear poly-p. The work is interesting.

However, I have some minor questions:

1. Authors did not used in-vivo system to clarify their effect of nano-particle

2. Aim was not elucidated clearly

3. “PolyP-NP bound 44 weaker to histone and caused more pronounced thickening of the fibrin fibers than linear polyp” ….please clarify it.

4. Authors used turbidimetric method in their experiment. In there any effect on platelet aggregation?

5. 0.005% Tween?

6. Authors may discuss the brief mechanism (2-3 sentences) in context of their nano-particle comparison-NET-histone-DNA, which was missing in conclusion section.

Reviewer #2: In this article, Lovas et al analyzed the fibrin polymerization from fibrinogen and clot lysis (plasmin) in presence of polyphosphate, polyphosphate nanoparticle NP, DNA and histone.

In the context of NETs and thrombosis, interplay between histones, DNA, fibrinogen are very crucial.

Author found that polyphosphate nanoparticle binding to histones are less than its linear conformers. Most probably, this is the reason why polyphosphate nanoparticle show less inhibition of polyphosphate nanoparticle-mediated inhibition of histone-dependent clot formation in comparison to the linear one. However, it would be difficult to conclude as DNA may have similar effects.

Overall, experiments were conducted logically, but still lack in details.

Major:

1. polyphosphate nanoparticle synthesis should be described elaborately.

2. Fig 1B should be labeled clearly to understand the 2 graph (112 & 188) are these 2 different molecules/preparation ??

3. What is the significance of stability (fig2 ). If it is not physiologically significant, need to be removed or transferred to suppl.

4. Figure 3A, 3B, 4A, 4B curve should be labeled. otherwise it is difficult to understand.

Presence of Ca in polyphosphate nanoparticle preparation may interfere the experiment. Linear polyphosphate experiments should contain similar Ca.

5. Why polyphosphate nanoparticle induce better clot formation and less clot lysis ? explanation should be added in discussion.

6.Table 3. histone concentration is too high. 0.5 to 5 uM concentration is clinically relevant and should be studied accordingly. Which type (CTH/H3/H4) of histone used ? Is all the histones have same effects ? need to be verified.

7. What is the effect of polyphosphate nanoparticle on thrombin activity, as polyphosphate can accelerate this. Moreover, TAFI can be activated in presence of polyphosphate. Is polyphosphate nanoparticle interfere TAFI activity (experiment or explanation required).

8. Polyphosphate and Histones interact and induce VWF from endothelial cells and platelets. VWF is one of the important factor for coagulation, inflammation and thrombosis. In the context of NETs, VWF plays a key role by interacting with fibrin, histones and DNA.

Author should mentioned these research.

6. PLOS authors have the option to publish the peer review history of their article (what does this mean?). If published, this will include your full peer review and any attached files.

Reviewer #1: No

Reviewer #2: **Yes: **Indranil Biswas

---

## [Author Response · Author response to Decision Letter 0]

5 Mar 2022

Reviewer #1: In this manuscript “ Polyphosphate nanoparticles enhance the fibrin stabilization by histones more efficiently than linear polyphosphates” Lovas et al demonstrated Polyphosphate nanoparticle with histone is the stronger modulator of fibrin formation than the linear poly-p. The work is interesting.

However, I have some minor questions:

Response: We would like to thank the reviewer for the careful evaluation of our manuscript. We are very pleased that the reviewer has found our “work is interesting”. At the same time, the reviewer pointed out some possible gaps in the presentation and discussion of the results in the manuscript, and suggested some improvements in this respect. We have addressed all the comments of the reviewer in detail below and revised our manuscript accordingly. We believe that this has helped us to improve and strengthen the manuscript. The changes to the manuscript are highlighted in red. Itemized responses to each of the comments are summarized below.

1. Authors did not used in-vivo system to clarify their effect of nano-particle

Response: We thank the reviewer for this comment, because it pointed out that we haven’t emphasized that the aim of the study was to develop a formulation of polyphosphates that adequately represents their in vivo particulate form (as documented by our DLS and TEM data), and is sufficiently stable to be used in repeated series of in vitro measurements (as applied by us in the study of fibrin structure and lysis as an example of an in vitro application). Accordingly we modified the formulation of the study aims (l. 64-68): “In this study, we aimed 1) to develop a new method for preparing stable, pure PolyP-NPs that are suitable for in vitro applications to study their role in hemostasis, and 2) to compare and characterize the effects of the linear and nanoparticulate forms of PolyPs and the combined effects of DNA, histones and PolyPs on the kinetics of clotting, structure and lytic susceptibility of fibrin clots in vitro.”

2. Aim was not elucidated clearly

Response: We agree that the aim was not sufficiently detailed in the abstract. Accordingly we expanded it focusing on the in vitro aspect of the study (l. 30-32). “Aims-To compare the effects of linear and nanoparticulate polyphosphates, and their combinations with relevant NET components (DNA, histone H3) on fibrin formation, structure, and lysis in in vitro assays focusing on histone-polyphosphate interactions.”

3. “PolyP-NP bound 44 weaker to histone and caused more pronounced thickening of the fibrin fibers than linear polyp” ….please clarify it.

Response: We agree that the referred last sentence in the Results section of the Abstract was confusing. Accordingly we have paraphrased this part of the Abstract for clarity (l. 42-45). “PolyP-NP, but not linear PolyP enhanced the prolongation of lysis time in fibrin containing histone and caused more pronounced thickening of the fibrin fibers than the linear form. Finally, PolyP-NP bound weaker to histone, than the linear form.”

4. Authors used turbidimetric method in their experiment. In there any effect on platelet aggregation?

Response: In this study we used a simplified thrombin-mediated fibrin clot model, in which we tested the polyphosphate effects in the absence of platelets. However we agree with the reviewer that the known direct and indirect effects of linear polyphosphates on platelet activation represent a relevant line for future studies, in which the nanoparticulate formulation developed by us can be applied. Accordingly we added the following text in the Discussion including 2 new references (l. 400-409, 411-414): ”Linear polyphosphates induce directly platelet aggregation and the long-chain variants are more potent in this action [28]. Linear polyphosphates modulate platelet activation indirectly, too. They induce von Willebrand factor release from endothelial cells through a high mobility group box 1 dependent mechanism resulting in platelet string formation [29]…. However, linear polyphosphates are unstable in circulation because of the presence of phosphatases. Thus, it is important to assess, if the particulate form of the polyphosphates that is more resistant to degradation can exert the same prothrombotic effects as their linear counterpart.”

5. 0.005% Tween?

Response: The text was corrected (l. 175): “Tween-20 at 0.005 w/w%. 

6. Authors may discuss the brief mechanism (2-3 sentences) in context of their nano-particle comparison-NET-histone-DNA, which was missing in conclusion section.

Response: We agree that a comment on the possible mechanism of the observed effects was missing in the conclusions section. Accordingly we have added the following text (l. 491-501): “We found that the PolyP-histone interaction is more significant than the PolyP-DNA interplay probably due to differences in molecular charges: opposite charges are accompanied by stronger interactions. In addition, PolyP nanoparticles enhance the thrombus stabilizing effects of histone more effectively than the linear form. The underlying mechanism of this difference could be related to the weaker PolyP-NP/histone binding observed by us. The weak binding of PolyP-NP to histones allows for independent action of these fibrin-stabilizing factors with consequent additivity of their effects on fibrin fiber thickening and lytic resistance. In contrast, the stronger linear PolyP-histone interaction counteracts more efficiently the accelerating effect of histones in clotting, but their complex formation abrogates the independence of their individual effects on fibrin structure and lysis and thus their combined action is not additive in the stabilization of fibrin.”

 

Reviewer #2: In this article, Lovas et al analyzed the fibrin polymerization from fibrinogen and clot lysis (plasmin) in presence of polyphosphate, polyphosphate nanoparticle NP, DNA and histone. In the context of NETs and thrombosis, interplay between histones, DNA, fibrinogen are very crucial. Author found that polyphosphate nanoparticle binding to histones are less than its linear conformers. Most probably, this is the reason why polyphosphate nanoparticle show less inhibition of polyphosphate nanoparticle-mediated inhibition of histone-dependent clot formation in comparison to the linear one. However, it would be difficult to conclude as DNA may have similar effects.

Overall, experiments were conducted logically, but still lack in details.

Response: We appreciate the positive comments about our work, including that it addresses “very crucial” research questions and that our “experiments were conducted logically”. We are grateful for the constructive comments and suggestions that we have addressed with revisions in the manuscript as itemized below.

Major:

1. polyphosphate nanoparticle synthesis should be described elaborately.

Response: In line with this suggestion, we added two sentences in the methods section so that it is apparent that the detailed recipe is there. The initial part of the section (l. 81-83) now reads: “Pure PolyP nanoparticles are labile, so we developed a novel technique for preparing stable PolyP nanoparticles based on improvements of earlier work [10,11]. The detailed procedure is as follows.” The final filtration step was also more detailed (l. 89-91) “Remnant large aggregates were removed by filtrating them sequentially through a syringe filter of 1.2 μm and then through a filter of 0.22 μm pore size. Finally, the filtrate with the remaining PolyP-NPs was retained and stored at 4 °C.”

2. Fig 1B should be labeled clearly to understand the 2 graph (112 & 188) are these 2 different molecules/preparation ??

Response: The figure legend has been changed in order to clarify the origin of the two curves in Fig. 1B (l. 215-217) “The distribution curves of two different PolyP-NP preparations are shown to illustrate the minimal and maximal median values of particle size in the PolyP-NP preparations used in this study.”

3. What is the significance of stability (fig2 ). If it is not physiologically significant, need to be removed or transferred to suppl.

Response: We agree with the reviewer that Fig. 2 has no physiological relevance, but it is essential to illustrate the major methodological improvement achieved with our study, because the pure PolyP-NP preparations used in previous studies were stable for several hours only, whereas ours preserved their structure for several days. That’s why we would like to retain the figure in the main body of the paper. We also agree that the original text did not explain clearly how we interpret the stability of the preparation. Now we’ve corrected this gap expanding the main text on l. 206-207 “no change in the median diameter of PolyP-NPs was observed suggesting that neither aggregation, nor disintegration of the PolyP-NPs occurred in the preparations” and the figure legend l. 219-221“The particle size of the PolyP-NPs was monitored by dynamic light scattering (DLS) for 6 days as a measure of the preparation stability”.

4. Figure 3A, 3B, 4A, 4B curve should be labeled. otherwise it is difficult to understand.

Presence of Ca in polyphosphate nanoparticle preparation may interfere the experiment. Linear polyphosphate experiments should contain similar Ca.

Response: The representative original curves in Figs. 3A&B, 4A&B have been labelled in the graphs and the legend was adapted accordingly.

We agree with the reviewer concerning the effect of Ca in polyphosphate nanoparticle preparations. That’s why in all cases we compare the nanoparticle effects to controls containing the same amount of Ca. However, the suggestion to have similar Ca in the linear polyphosphate experiments is not feasible technically, because if Ca is added to the linear PolyP sample, it would precipitate in an uncontrolled manner, therefore it would not be a linear PolyP anymore. That’s why all control measurements for the effects of linear PolyP are performed in Ca-free medium.

5. Why polyphosphate nanoparticle induce better clot formation and less clot lysis ? explanation should be added in discussion.

Response: We agree with the reviewer that in the original Discussion we referred to our previous work on the fibrin structure/lysibility interrelations in the presence of histones alone or in combination with linear PolyP (Refs. 2, 5), but we didn’t provide any explanation for the difference in the effects of the two forms of PolyP. To address this point we have added the following text in the Discussion (l. 493-501): “In addition, PolyP nanoparticles enhance the thrombus stabilizing effects of histone more effectively than the linear form. The underlying mechanism of this difference could be related to the weaker PolyP-NP/histone binding observed by us. The weak binding of PolyP-NP to histones allows for independent action of these fibrin-stabilizing factors with consequent additivity of their effects on fibrin fiber thickening and lytic resistance. In contrast, the stronger linear PolyP-histone interaction counteracts more efficiently the accelerating effect of histones in clotting, but their complex formation abrogates the independence of their individual effects on fibrin structure and lysis and thus their combined action is not additive in the stabilization of fibrin.”

6.Table 3. histone concentration is too high. 0.5 to 5 uM concentration is clinically relevant and should be studied accordingly. Which type (CTH/H3/H4) of histone used ? Is all the histones have same effects ? need to be verified.

Response: We agree with the reviewer that histone concentrations up to 5 µM have been reported in circulating blood (for example in baboon sepsis model, Xu et al, Nat. Med. 2009; 15, 1318). However, when NETs are formed in thrombi the local concentration of histones is expected to be at least an order of magnitude higher. That’s why in studies of the role of NETs in thrombi histone concentrations up to 1,000 µg/mL (about 70 µM) are considered relevant (for example, Fuchs et al, Blood, 2011;118:3708). Thus, the concentration range chosen by us in Table 3 starts with a value relevant for the circulation and goes up to values that are typical for the local environment of the forming NETs. Concerning the type of histone used in our study we indicate in the Materials section (l. 73) that we used “histones from calf thymus type VIIIS (arginine-rich histone H3) “. We chose this type, because in our previous work (Refs. 2 and 5) we didn’t find any significant difference between the effects of the linker H1 and the core H3 histones on fibrin structure and lysis (as we indicated in Ref. 5). So we performed the extensive series of measurements reported now only with the H3 histone.

7. What is the effect of polyphosphate nanoparticle on thrombin activity, as polyphosphate can accelerate this. Moreover, TAFI can be activated in presence of polyphosphate. Is polyphosphate nanoparticle interfere TAFI activity (experiment or explanation required).

Response: In our study we used purified fibrinogen and monitored the kinetics of its conversion to fibrin by thrombin, as illustrated in Figs. 3A and 4A. Because fibrinogen is the natural substrate of thrombin, we think that the clotting time data reported in Figs. 3C and 4C are adequate indicators of the changes in thrombin activity. We agree that the known effects of linear polyphosphates on other hemostatic/fibrinolytic factors present in plasma prompt the evaluation of the nanoparticulate variants on the same factors, which could be a future application of the stable nanoparticles reported now. Accordingly we have expanded the Discussion section placing the utility of our polyphosphate formulation in the context of studies targeting other plasma factors, but emphasizing that the current work focused on a simplified pure fibrin experimental setup (l. 396-413): “Multifaceted interactions of linear polyphosphates with thrombogenic factors have been observed. Linear polyphosphates accelerate blood clotting by activating the contact pathway and activation of factor V, which counteracts the action of a major anticoagulant protein, tissue factor pathway inhibitor [27]. They also delay clot lysis through enhancing the action of thrombin-activatable fibrinolysis inhibitor [27]….. However, linear polyphosphates are unstable in circulation because of the presence of phosphatases. Thus, it is important to assess, if the particulate form of the polyphosphates that is more resistant to degradation can exert the same prothrombotic effects as their linear counterpart. To this end we produced a stable suspension of polyphosphate nanoparticles and in the present study we focused on one aspect of the multifaceted role of polyphosphates in thrombus formation, structure and lysis. We compared the effects of linear and nanoparticulate polyphosphates alone and in combination with NET components (DNA, histone) utilizing a simplified thrombin-mediated fibrin clot model.”

8. Polyphosphate and Histones interact and induce VWF from endothelial cells and platelets. VWF is one of the important factor for coagulation, inflammation and thrombosis. In the context of NETs, VWF plays a key role by interacting with fibrin, histones and DNA.

Author should mentioned these research.

Response: We agree with the reviewer that modulation of platelet function via VWF is an essential aspect of the polyphosphate effects in haemostasis. Accordingly we added the following text in the Discussion including 2 new references (l. 400-404): ”Linear polyphosphates induce directly platelet aggregation and the long-chain variants are more potent in this action [28]. Linear polyphosphates modulate platelet activation indirectly, too. They induce von Willebrand factor release from endothelial cells through a high mobility group box 1 dependent mechanism resulting in platelet string formation [29]”.

---

## [Decision Letter · Decision Letter 1]

28 Mar 2022

Polyphosphate nanoparticles enhance the fibrin stabilization by histones more efficiently than linear polyphosphates

PONE-D-21-40112R1

Dear Dr. Kolev

We’re pleased to inform you that your manuscript has been judged scientifically suitable for publication and will be formally accepted for publication once it meets all outstanding technical requirements.

Kind regards,

Gausal A. Khan, Ph.D;CSci,FRSB

Academic Editor

PLOS ONE

Additional Editor Comments (optional):

Reviewers' comments:

Reviewer's Responses to Questions

**Comments to the Author**

1. If the authors have adequately addressed your comments raised in a previous round of review and you feel that this manuscript is now acceptable for publication, you may indicate that here to bypass the “Comments to the Author” section, enter your conflict of interest statement in the “Confidential to Editor” section, and submit your "Accept" recommendation.

Reviewer #1: All comments have been addressed

Reviewer #2: All comments have been addressed

2. Is the manuscript technically sound, and do the data support the conclusions?

Reviewer #1: Yes

Reviewer #2: Yes

3. Has the statistical analysis been performed appropriately and rigorously? 

Reviewer #1: Yes

Reviewer #2: Yes

4. Have the authors made all data underlying the findings in their manuscript fully available?

Reviewer #1: (No Response)

Reviewer #2: Yes

5. Is the manuscript presented in an intelligible fashion and written in standard English?

Reviewer #1: Yes

Reviewer #2: Yes

6. Review Comments to the Author

Reviewer #1: (No Response)

Reviewer #2: Authors have addressed the reviewers suggestion and concern. Revised manuscript is improved in explanation of the experimental evidence.

7. PLOS authors have the option to publish the peer review history of their article (what does this mean?). If published, this will include your full peer review and any attached files.

Reviewer #1: No

Reviewer #2: **Yes: **Indranil Biswas

---

## [Editor Report · Acceptance letter]

6 Apr 2022

PONE-D-21-40112R1 

Polyphosphate nanoparticles enhance the fibrin stabilization by histones more efficiently than linear polyphosphates 

Dear Dr. Kolev:

I'm pleased to inform you that your manuscript has been deemed suitable for publication in PLOS ONE. Congratulations! Your manuscript is now with our production department. 

Kind regards, 

on behalf of

Prof. Gausal Azam Khan 

Academic Editor

PLOS ONE